# Long-Term Memory Formation in *Drosophila* Depends on the 3′UTR of CPEB Gene *orb2*

**DOI:** 10.3390/cells12020318

**Published:** 2023-01-14

**Authors:** Eugene N. Kozlov, Elena V. Tokmatcheva, Anastasia M. Khrustaleva, Eugene S. Grebenshchikov, Roman V. Deev, Rudolf A. Gilmutdinov, Lyubov A. Lebedeva, Mariya Zhukova, Elena V. Savvateeva-Popova, Paul Schedl, Yulii V. Shidlovskii

**Affiliations:** 1Laboratory of Gene Expression Regulation in Development, Institute of Gene Biology, Russian Academy of Sciences, 119334 Moscow, Russia; 2Institute of Physiology, Russian Academy of Sciences, 188680 St. Petersburg, Russia; 3Center for Precision Genome Editing and Genetic Technologies for Biomedicine, Institute of Gene Biology, Russian Academy of Sciences, 119334 Moscow, Russia; 4Department of Biology and General Genetics, Sechenov First Moscow State Medical University (Sechenov University), 119992 Moscow, Russia; 5Department of Molecular Biology, Princeton University, Princeton University, Princeton, NJ 08544-1014, USA

**Keywords:** 3′untranslated region, CPEB proteins, long- term memory, synapse, protein localization

## Abstract

Activation of local translation in neurites in response to stimulation is an important step in the formation of long-term memory (LTM). CPEB proteins are a family of translation factors involved in LTM formation. The *Drosophila* CPEB protein Orb2 plays an important role in the development and function of the nervous system. Mutations of the coding region of the *orb2* gene have previously been shown to impair LTM formation. We found that a deletion of the 3’UTR of the *orb2* gene similarly results in loss of LTM in *Drosophila.* As a result of the deletion, the content of the Orb2 protein remained the same in the neuron soma, but significantly decreased in synapses. Using RNA immunoprecipitation followed by high-throughput sequencing, we detected more than 6000 potential Orb2 mRNA targets expressed in the *Drosophila* brain. Importantly, deletion of the 3′UTR of *orb2* mRNA also affected the localization of the Csp, Pyd, and Eya proteins, which are encoded by putative mRNA targets of Orb2. Therefore, the 3′UTR of the *orb2* mRNA is important for the proper localization of Orb2 and other proteins in synapses of neurons and the brain as a whole, providing a molecular basis for LTM formation.

## 1. Introduction

Synaptic plasticity is one of the key mechanisms underlying learning and memory formation. The mechanism is based on the ability of neurons to alter the strength and efficiency of signal transmission through synapses in response to stimulation [1]. Transmission of the signal can be amplified (a process of potentiation) or weakened (a process of depression), and changes in signal transmission can be of short duration or can persist for a longer period [2]. Short-term alterations in synaptic function and the formation of short-term memory can be induced without protein or mRNA synthesis, while long-lasting alterations in synapse function and the formation of long-term memory (LTM) require mRNA and protein syntheses [3]. It is thought that mRNA and protein syntheses alter the functional properties of individual synapses or induce new synaptic connections. However, these changes in synaptic function are most likely not restricted to a single synapse, but rather occur throughout a network of neurons that are responsible for capturing memory [4]. 

One of the mechanisms that alter local synaptic functionality and/or induce de novo assembly of new synaptic junctions is based on local translation and synthesis of proteins involved in maintaining synaptic junctions. The process requires transport of mRNAs important for the synaptic function from the cell body to synaptic junctions [5]. In the absence of stimulation, at least a subset of these mRNAs should be stored in a translation-repressed form, while some mechanisms should induce their local translation upon stimulation. Moreover, changes induced in synaptic function by de novo translation should be maintained in order to sustain memory [6].

Cytoplasmic polyadenylation element (CPE)-binding (CPEB) proteins are a family of translation factors involved in LTM formation [7,8]. The CPEB proteins bind to U-rich sequences known as CPEs in the 3′UTRs of mRNAs to control their translation [9]. The CPEB proteins are widespread among eukaryotes and have prion-like properties They are capable of forming multimeric complexes of amyloid structure, which perform physiological functions in contrast to pathogenic amyloids [10,11]. Depending on the biological context and regulatory input, CPEB proteins can activate or repress translation [12].

*Drosophila* has two CPEB family proteins, Orb and Orb2. Orb is expressed almost exclusively in female germline cells and plays a critical role in oocyte specification and development [13,14,15]. However, Orb is also found in a subset of neurons in mushroom bodies and participates in LTM formation [16].

Orb2 is widely expressed in somatic cells during embryogenesis, male germ cells of adult flies, and the nervous system [17,18]. There are two Orb2 isoforms, Orb2A and Orb2B. Both isoforms have polyglutamine (polyQ), RNA-binding, and zinc finger domains, but differ in their N termini [7]. Orb2B is widely distributed in the *Drosophila* brain, while Orb2A is thought to be largely restricted to the synaptic zone of the mushroom body neurons [8,19]. Both of the Orb2 isoforms are essential for the formation of LTM; however, they are believed to have different functions, because the polyQ domain of Orb2A and the RNA-binding domain of Orb2B are essential for LTM. It is believed that synaptic stimulation induces a conformational change in the polyQ domain of Orb2A and that the change leads initially to self-aggregation of Orb2A and further aggregation of Orb2A and Orb2B in heteromultimeric complexes. In this model, memory is maintained by self-induction of aggregation of newly synthesized Orb2A and Orb2B [20]. It has also been suggested that a transition from the unaggregated to the aggregated state alters Orb2 activity. Monomeric Orb2 represses mRNAs localized in synapses, while aggregated Orb2 functions as a translational activator [21].

Neurons differ from most other somatic cells in that many of their mRNAs have extended 3′UTR sequences [22,23]. It is thought that these neuron-specific 3′UTR isoforms contain regulatory elements that are important for mRNA transport, translational regulation, and stability [24]. The *orb2* gene also has multiple mRNA isoforms (Figure 1A), and five of them (RA, RB, RC, RD, RH) have been described. RA is the shortest mRNA, encodes Orb2A, and has 3′UTR of only 395 nt [17]. The other *orb2* mRNAs encode Orb2B and carry 3′UTRs ranging in size from 583 to 5794 nt. The RH isoform contains 37 CPE motifs (22 canonical A-containing sites and 15 noncanonical G-containing sites [9,25]), while the RD isoform has 27. In contrast, the RA transcript has no CPEs, while RB has two and RC has seven. The large number of CPEs in the different *orb2* mRNAs indicates that the Orb2 protein is likely capable of regulating the localization and translation of its own mRNA [18,26].

The very large number of CPEs in the two longest *orb2* mRNA isoforms and the presence of these large mRNAs in neurons prompted us to investigate the functions of the *orb2* 3′UTR in neurons. For this purpose, we used the 4522-bp 3′UTR deletion, *orb2^R^*, which was generated previously [26]. The deletion extends from near the 3′ end of the *orb2* RC sequence to just upstream of the RH polyadenylation site and removes all but five of the *orb2* CPEs. The remaining CPEs are common to the RC, RD, and RH mRNAs. Here we examined the phenotypic effects of the deletion on the nervous system in adult flies. 

## 2. Materials and Methods

### 2.1. Fly Strains

The following *Drosophila* fly strains were used in this study: a strain with a 3′UTR deletion of the *orb2* gene (*orb2^R^* [26]), which was obtained using the CRISPR/Cas9 system; the parental Cas9-expressing fly strain *51324* (*w[1118]; PBacVK00027*) from the Drosophila Bloomington stock center (DBSC, USA) was used as a control to measure the mRNA and protein levels and to perform whole mount staining; learning and memory formation are similar in *51324* and WT Canton S (*CS*) flies, and the *CS* strain was therefore used as a control strain in behavioral experiments; a strain that expressed the Orb2-GFP fusion instead of the WT *orb2* allele (a kindly gift from Kristina Keleman) was used in immunoprecipitation experiments.

### 2.2. Locomotion Assays

In a climbing assay, 10 flies from each strain (WT and *orb2^R^*) were placed in a standard empty Drosophila-culture vial with a marked 6 cm distance from the bottom. Flies were knocked to the bottom of vial with series of 10 strong hits on the table. The number of flies that climbed 6 cm in 10 s was counted. A total of 100 flies of each genotype were tested and the percentage of flies that climbed up in the established time was calculated. In a bang-sensitivity assay, 10 flies were placed in an empty plastic vial and shaken on an microspin FV-2400 (500 g) for 10 s, and those that returned to a standing position in 10 and 20 s were counted. A total of 100 flies of each genotype were tested and the percentage of flies that climbed up in the established time was calculated. In a heat-shock assay, flies grown at room temperature were placed in an empty preheated vial and heat shocked at 37 °C in cultural thermostat for 10 min. The number of flies that were still standing after a 10-min heat shock was counted.

### 2.3. Behavioral Assay

A courtship-suppression paradigm was used to check memory formation [27]. Flies were raised on standard *Drosophila* medium (cornmeal, agar, molasses, yeast, and nipagin) at 25 °C and 60–70% humidity with a 12-h light–dark cycle. Males of the strain under study were collected without anesthesia 0–10 h after eclosion and kept separately in small food vials until the tests were carried out. Training was performed using 5-day-old *CS* females fertilized 1 day before in the experimental chamber (15 mm in diameter, 5 mm in height). Learning and testing were performed at an age of 5 days. For training, males without experience in courtship behavior were placed in a vial with a fertilized female for 30 min. For LTM formation, male and female flies were kept in the same vial for 5 h.

Short-term memory was assessed immediately after training, and mid-term memory, after 180 min. LTM was assessed 2 days after training, using newly fertilized 5-day-old *CS* females. Males without courtship behavior experience were used as a control. In each group, 20 males were tested. Behavior of males was assessed immediately, 3 h, and 2 days after learning (30-min training). The time spent in courtship (orientation, following, wing vibration, licking and attempted copulation) was recorded for 300 s when tested males were placed in an experimental chamber with a previously mated female. The courtship index (percentage of time spent in courtship) was calculated for each male. A courtship index (CI) was calculated as a ratio of the male courtship time to the entire observation time [27]. A learning index (LI) was calculated as LI = [(CIn − CIexp)/CIn] × 100%, where CIn is the average CI of two independent samples of males without courtship behavior experience, and CIexp is the average CI of two independent samples of males after training. Statistical comparisons of behavioral data were made by a randomization test, by directly calculating the probability of rejection of the null hypothesis α_R_. A sampled randomization test with 10,000 permutations was used. The null hypothesis was rejected at α_R_ < 0.05. A two-sided test was used to compare the courtship indices. A one-sided test was used to compare learning indices.

### 2.4. Antibodies

We used the following antibodies against Orb2: 4G8 (1:50 dilution) for Western blotting and 2D11 for whole brain staining (1:100 dilution). The antibodies were deposited in the Developmental Studies Hybridoma Bank (DSHB) by P. Schedl. Anti-GFP antibodies DSHB-GFP-2G6, 4C9, 8H11, and 12A6 were deposited in the DSHB by DSHB investigators (DSHB Hybridoma Product DSHB-GFP-2G6, DSHB-GFP-4C9, DSHB-GFP-8H11 and DSHB-GFP-12A6). An antibody against the eyes absent (Eya) protein (10H6) was deposited in the DSHB by S. Benzer and N.M. Bonini. The DCSP-1 (ab49) antibody against the cysteine-string protein (CSP) was deposited in the DSHB by E. Buchner and A. Hofbauer. An anti-polychaetoid (PYD1) antibody was deposited in the DSHB by A.S. Fanning. Antibodies were used at a 1:500 dilution in Western blotting and a 1:200 dilution in whole brain staining. The 3C11 antibody against Synapsin was deposited in the DSHB by B.R.E. Klagge and colleagues and used at 1:200 dilution. The ADL67.10 antibody against the lamina protein (Adl) was deposited in the DSHB by P.A. Fisher and was used at a 1:1000 dilution in Western blotting. Alexa 488-conjugated goat anti-mouse IgG antibody (Invitrogen, USA, catalog number A-11001) was used as a secondary antibody at a 1:1500 dilution. HRP-conjugated goat anti-mouse IgG antibody (Jackson ImmunoResearch, UK, catalog number 115-035-174) was used as a secondary antibody in Western blotting at 1:2000.

### 2.5. RNA Immunoprecipitation

To collect an appropriate amount of fly heads, about 10 g of flies of the Orb2-GFP strain were frozen in liquid nitrogen, vortexed, and sifted through two 800- and 380-µm sieves to separate heads from bodies. All the following steps were carried out at 4 °C. Using a Teflon pestle, heads were homogenized in a polysome buffer (10 mM HEPES, pH 7.4, 100 mM KCl, 5 mM MgCl_2_, 0.5% NP-40, 1 mM DTT) supplemented with a protease-inhibitor cocktail (Roche, catalog number 11836145001) and a ribonuclease inhibitor (Ribolock, Thermo Fisher Scientific, catalog number EO0382). The lysate was centrifuged twice at 1000× *g* for 5 min to pellet the debris. Of the final volume (2 mL), 100 µL of the clarified lysate was used for RNA isolation and 10 µL, for Western blot analysis. The samples were designated as Input.

Further, the initial lysate was incubated with an anti-GFP antibody mix (4C9, 8H11, 2G6, and 12A6 with a concentration of 30 ng/µL each) for 24 h at +4 °C. A negative control sample contained immunoglobulin G (IgG) from a pre-immune serum. After incubation, 10 µL of protein G Sepharose (Invitrogen, catalog number 101241) was added to each sample, incubated for 2 h at room temperature, and washed three times with the polysome buffer for 15 min. The resulting Sepharose was resuspended in 60 µL of the polysome buffer; 2 µL was used in Western blot analysis, and the rest of the suspension was used for RNA isolation to assess the quality of the precipitation reaction.

### 2.6. RNA Isolation, Reverse Transcription, qPCR

To isolate RNA, we used Trizol reagent according to the manufacturer’s recommendations (Molecular Research Center, USA, catalog number TR118). The final RNA concentration was measured with a RNA assay kit (Molecular probes, USA, catalog number Q32852). Equal amounts of Input RNA, immunoprecipitate (IP) obtained with anti-GFP antibodies, and mock IP were used to synthesize cDNA with a RevertAid kit (Invitrogene). The resulting cDNA was used for qPCR in three replicates of each sample. To assess the quality of the IP reaction, the enrichment of Orb2 target mRNAs was measured using primers *act5c, Orb2, tub56D* (positive control), 28S, and *RplP0* (negative control) RNAs. Five independent reaction sets were tested, and the most enriched IP sample obtained with anti-GFP antibodies was used for high-throughput sequencing.

To measure the *orb2* mRNA level, extracts from 20 fly heads of the *51324* and *Orb2^R^* strains were used to isolate RNA; cDNA was synthesized, and qPCR was carried out with primers to *Orb2* and *gapdh2* as a reference gene. We performed five independent biological replicates and used Student’s *t*-test (*p* < 0.05).

The following primers were used in qPCR: *act5C* (5′-GGCACCACACCTTCTACAATGAGC and 5′-GAGGCGTACAGCGAGAGCACAG); *tub56D* (5′-CGAGAACACGGACGAGACCTACTG and 5′-GGAATCGGAGGCAGGTGGTTACG); *orb2* (5′-TAACACCAGCGAAAGGGGAC and 5′-TCAGATGTGCGACGAGTGC); *RplP0* (5′-ATCAAGGTTGTGGAACTGTTCG and 5′-GCGGGTTGTTCTCCAGATGA); *gapdh2* (5′-CTACCTGTTCAAGTTCGATTCGAC and 5′-AGTGGACTCCACGATGTATTCG); 2and 8S RNA (5′-AATTCAGAACTGGCACGGACTTGG and 5′-AGAGCACTGGGCAGAAATCACATTG).

### 2.7. Next-Generation Sequencing Data Analysis

RNA isolation and preparation of cDNA libraries from Input, IP with antibodies, and mock IP were performed using a NEB E7770S kit (New England Biolabs, USA) according to the manufacturer’s recommendations. The cDNA libraries were sequenced on the Illumina 2000 platform. Reads were mapped to the *Drosophila* genome using TopHat software [28]. Next, the number of reads that mapped to exons was calculated using bedtools followed with htseq-count [29] calculation of reads for each gene; this value was used to calculate CPM (counts per million). Raw reads were deposited in the BioProject database with accession No. PRJNA899720 (SRA accessions are SRR22244251, SRR22245477, and SRR22254174). GO-term analysis was performed by the service PANTHER [30]. Enrichment by published gene sets was performed using the service FlyMine [31].

### 2.8. Crude Synaptosome Preparation

For each genotype, 20 fly heads were homogenized in 50 µL of 0.32 M sucrose buffer (5 mM HEPES, pH 7.4, 145 mM NaCl, 5 mM KCl, 2 mM CaCl_2_, 1 mM MgCl_2_, 5 mM glucose) supplemented with a protease inhibitor cocktail. Crude synaptosome preparations were obtained essentially as described in [32] with an extra centrifugation step added after collecting the first pellet.

### 2.9. Semiquantitative Western Blot Analysis

To assess the distribution of the proteins of interest in *Drosophila* nervous cells, protein concentrations in a cell soma fraction and a synaptosome fraction were measured using a Qubit protein assay kit (Molecular probes, catalog number Q33211). Equal amounts of each sample were diluted by factors of 2, 4, and 6 and loaded onto precast stain-free PAAG (Biorad, USA, cat. number 456-8095); bands were visualized in a Chemidoc system to quantify the protein concentration. Chemiluminescence was measured with the same system.

### 2.10. Whole Mount Immunostaining

The brains of 5-day-old flies were dissected, washed twice with ice-cold PBS supplemented with 0.05% Triton-X100 (PBSt), fixed with methanol for 20 min, and treated with acetone for 5 min for permeabilization. After two wash steps, the brains were incubated in 4% skim milk with 1% BSA overnight (+4 °C). A primary antibody was added, and incubation continued for 24 h with rotation (12 rpm) at +4 °C. After 3× washing with PBSt for 30 min each, a secondary antibody was added, and incubation was continued for 2 h at room temperature with rotation (12 rpm). After washing with PBSt for 1 h, the samples were placed in a mounting medium with DAPI (Vector Labs, USA). Images were obtained using a Leica STELLARIS 5 confocal microscope. The stained samples were scanned in multichannel mode using a 40× oil objective with numerical aperture 1.3. Images with a frame size of 2048 × 2048 pixels and a z resolution of 1 µm were taken at a scan speed of 400. At least twenty brain images were obtained for each genotype, and the selected images were processed using LazX software (Leica Application Suite X, version 3.7.25997.6., Leica Microsystems, Germany)

## 3. Results

### 3.1. Deletion of orb2 3’UTR Does Not Affect Locomotion

The Orb2 protein is widely distributed in the *Drosophila* brain, including the ellipsoid and fan-shaped body structures [17], which are responsible for walking, assessing distance to an object, and orientation in space. In previous studies, a *piggyBac* transposon inserted into the first intron of the *orb2* gene (*orb2^6090^*) was found to significantly decrease the Orb2 protein levels and to impair the ability of a fly to orient in space and to walk [17]. To assess the role of the extended *orb2* 3′UTRs in motor activity, we used the *orb2^R^*-deletion mutant generated previously. As described above, the mutation deletes a 4.5-kb DNA sequence encoding the larger *orb2B* mRNA isoforms and leaves the *orb2B* 3′UTR sequence that includes only five CPEs [26] (Figure 1A). Flies were tested for the abilities to climb after a strong impact (climbing assay), to restore orientation in space after intense shaking (vortex assay), and to stand up after heat shock.

Assessment of physical activity using the climbing assay showed an insignificant decrease in the speed of movement of *orb2^R^* flies after impact (Figure 1B). We did not find significant differences in the ability of *orb2^R^* flies to restore the body position after vortexing (Figure 1C). Likewise, the mutant flies were as resistant as WT flies to a brief heat treatment (10 min at 37 °C). Thus, *orb2^R^* flies clearly differed from flies carrying the null allele *orb2^6090^ piggyBac* in these assays.

### 3.2. Long-Term, but Not Short- or Mid-Term, Memory Formation Is Disrupted in the orb2^R^ Mutant

To investigate the effects of *orb2^R^* on learning and memory retention, we used the conditioned courtship suppression paradigm in males, which was tested after interacting with non-receptive females [27]. The training session was 30 min in the short- and mid-term memory assays. The behavior of males was assessed immediately after training to check short-term memory and 3 h later to check mid-term memory. We found no significant difference in courtship index between *orb2^R^* and WT males in the case of short- and mid-term memory (Figure 1D). The results are consistent with published data that a complete deletion of the *orb2* gene, a deletion of Orb2A (*orb2^ΔA^*), or a deletion of the polyglutamine domain of the Orb2 protein (*orb2^ΔQ^*) have no effect on the formation of short- or mid-term memory [19]. 

To assess LTM formation, male flies were first trained for 5 h and the courtship index was measured two days later. As shown in Figure 1E, *orb2^R^* males had a pronounced deficit in LTM; their courtship index was comparable to that of naïve untrained WT males. Thus, the effects of the 3′UTR deletion are similar to those observed for the *orb2^ΔA^* and *orb2^ΔQ^* mutations [7,19]: no alteration occurred in short-term or mid-term memory, but LTM was strongly impaired.

### 3.3. Orb2 Protein Levels in the Brain Are Altered in orb2^R^

While transcription of the *orb2* gene is not likely to be affected by the 3′UTR deletion, there could be alterations in the yield of fully processed *orb2B* mRNAs or changes in their stability. To test these possibilities, we used RT–qPCR with primers corresponding to an exon common to all *orb2* mRNA isoforms. The *gapdh2* mRNA was used as a reference. As shown in Figure 2A, the level of the *orb2* mRNA extracted from the heads of 5-day-old homozygous *orb2^R^* flies did not differ from that of WT flies.

The levels of the Orb2B protein were examined next. Note that Orb2A is not detectable in Western blots of fly heads [17]. Total lysates from the heads of 5-day-old WT and *orb2^R^* flies were analyzed by Western blotting with an anti-Orb2 antibody. To assess the relative amount of the Orb2B protein, we normalized the chemiluminescent signal detected by Western blotting to the total protein level in the sample. The Orb2B level in *orb2^R^* was about 2/3 of that in WT (Figure 2B,C). This finding suggests that the 3′UTR deletion reduces the stability or translation efficiency of the *orb2B* mRNAs.

### 3.4. Orb2 Accumulation in Synaptic Fractions Is Affected in orb2^R^ Mutant

A modest reduction in the amount of the Orb2 protein in head lysates seemed inconsistent with the clear deficit in LTM formation. Because 3′UTR sequences are believed to be important not only for translational regulation, but also for the mRNA localization in different subcellular compartments, we wondered if the distribution of the Orb2 protein within neurons is altered in mutant flies. To explore this possibility, we fractionated the total-head homogenate into neuronal cell body and synaptic fractions using procedures described previously [32]. A total-head homogenate (L) was first centrifuged through a 0.32 M sucrose solution to obtain a low-speed pellet fraction (LP) enriched in neuronal cell bodies (soma) (Figure 3A). The supernatant was then centrifuged through a 0.8 M sucrose solution. The resulting pellet fraction (P1) contained proteins from neuronal synapses together with a mixture of cytosolic proteins, while other cytosolic proteins were left in the supernatant (S1). The pellet was resuspended in a 0.8 M sucrose buffer and centrifuged again through a 0.8 M sucrose solution. The pellet fraction (P2) contained mostly synaptic proteins. To assess the purity of the synaptic fractions, we used antibodies against Lamin of the nuclear envelope and the cysteine string protein (Csp), which is involved in synaptic transmission [33]. Western blot analysis of the fractions showed that Lamin was present in the LP fraction but absent in the final synaptic fraction P2. On the contrary, Csp was enriched in both the P1 and P2 synaptic fractions (Figure 3A).

The Orb2B distribution through the fractions was compared between extracts prepared from WT and *orb2^R^* flies. For this purpose, the chemiluminescent signals from anti-Orb2 antibody staining were normalized to the total protein levels for each fraction. Orb2 protein levels in soma were similar in *orb2^R^* and WT flies (Figure 3B). At the same time, the Orb2B protein in synapses of *orb2^R^* flies was almost three times less abundant than in WT.

To confirm that the distribution of the Orb2 protein in neurons was altered in the *orb2^R^* mutant, we probed the brain whole mounts of WT and *orb2^R^* with an anti-Orb2 antibody. Figure 4A shows that there was a general decrease in immunofluorescent signal in the *orb2^R^* samples. Previously, the Orb2 protein has been shown to occur both in the perinuclear space (rings around nuclei) and axons of neurons in the fly brain [19]. In agreement with this, we found Orb2 in the neuron bodies and the neuropile region in WT (marked in Figure 4A,B). In *orb2^R^* samples, the Orb2 protein was detected predominantly in the vicinity of neuronal nuclei, forming a characteristic ring distribution (Figure 4C). The intensity of perinuclear staining was comparable between WT and *orb2^R^*, while the signal from neuropile region was significantly reduced in *orb2^R^* flies (Figure 4A,B).

Therefore, both biochemical and immunostaining approaches indicate that the amount of Orb2B localized in axons and synapses was decreased in the *orb2^R^* mutant. The abnormalities observed in Orb2B localization could potentially account for LTM impairment in *orb2^R^* flies. 

### 3.5. Identification of Orb2 mRNA Targets in the Nervous System

A reduction of the amount of Orb2B localized in the synaptic zone would be expected to impact the localization and/or translation of mRNAs that are normally located in this subcellular compartment. To identify the potential neuronal mRNA targets that might be impacted by a reduction of Orb2B protein levels, we used RNA immunoprecipitation followed by next-generation sequencing. Because the available anti-Orb2 antibodies efficiently precipitated Orb2 from head extracts, we used a fly strain in which the *orb2* gene encodes an Orb2-GFP fusion protein. It has been shown that *orb2-gfp* flies are fully viable and fertile and have normal LTM memory [19]. Antibodies against GFP were used to pull down Orb2-GFP and associated target mRNAs. The anti-GFP antibodies ensured a significant enrichment of the Orb2-GFP fusion protein in the experimental IP, while the Orb2-GFP protein was not detected in a mock IP (obtained with IgG of preimmune serum). We also found that only 10–15% of the Orb2-GFP protein was left in the output fraction after precipitation. 

We assessed the efficiency of precipitation of Orb2 target mRNAs (*act5C, orb2, tub56D* transcripts) using qPCR. The targets showed moderate excess of presence in IP obtained with the anti-GFP antibodies as compared to the initial lysate, while lower levels were observed in the mock IP. Negative control transcripts (28S rRNA and *RpLP0* mRNA) showed a similar precipitation efficiency in both IP with the anti-GFP antibodies and mock IP (Figure 5B). Thus, our protocol showed the efficiency and selectivity of the precipitation of target mRNAs. 

The sample of RNA coprecipitated with the anti-GFP antibodies was further analyzed by high-throughput sequencing. Reads were mapped onto the genome, and protein-coding genes whose transcripts were detected in both Input and IP samples were left in the list. The enrichment of each item was checked by two parameters, a ratio of CPM values in experimental IP to CPM in Input and a ratio of CPM in experimental IP to CPM in mock IP. We discarded genes with enrichment values lower than those observed for the *orb2, act5C* and *tub56D* mRNAs. Thus, we obtained a list of 6415 genes (Appendix A).

We compared our list with those described previously. Among 5298 targets found in S2 cells [34], about 60% were also present in our list. Among 27 Orb2 targets found in the nervous system [35], 21 were detected. The list included the *orb* and *apkc* genes, whose mRNAs have previously been identified as Orb2 targets [18,36].

Many genes on the list are specific to the nervous system; about 40% of the genes show moderately high levels of expression in fly heads [37]. Analysis of GO terms of the set showed enrichment of genes that encode proteins of axons and synapses; proteins of neuron bodies were also present. Genes of the list are involved in multiple biological processes related to nervous system functions, including synapse assembly and activity, learning, and memory (Figure 5C,D). Thus, we generated a list of putative Orb2 target mRNAs, which could be used for further analysis.

### 3.6. Intracellular Distribution of Proteins in Neurons Is Altered in orb2^R^ Flies

A plausible cause of LTM loss in *orb2^R^* flies is dysregulated translation of mRNAs in synapses because of the depletion and improper localization of the Orb2 protein. To explore this possibility, we selected genes whose mRNAs were recovered as potential *orb2* targets in our RNA-IP experiment and had been implicated in LTM formation. We chose *cysteine string protein* (*csp)* [38], *eyes absent* (*eya*) [39], and *polychaetoid* (*pyd*) [40]. 

First, we estimated the distribution of these proteins in synapses and the soma using a biochemical approach. The Csp protein was detectable in both somatic and synaptic fractions of WT and *orb2^R^* flies. Although the Csp amount in the soma relative to the total protein in *orb2^R^* flies was similar to that in WT flies, this was not the case with the synaptic fraction. As shown in Figure 6A, the Csp level in the synaptic fraction was nearly doubled in *orb2^R^* compared to WT. In the case of *eya,* we observed an approximately two-fold reduction in the amount of the Eya protein in the soma fraction, while a very limited amount of the Eya protein in the synaptic fraction was unchanged (Figure 6A). In the case of the Pyd protein, we detected a slight decrease in protein amount in neuron bodies, while there was a marked (~three-fold) increase in the relative amount of the protein in the synaptic fraction.

Next, we performed immunostaining of whole mount *Drosophila* brain samples with antibodies against these proteins. Staining with the antibody against Eya showed that Eya was found in nuclei and neuronal cell bodies in all parts of the central and lateral brain and the optic lobes. There was no overall change in Eya distribution in the mutant; however, as expected from the Western results, the signal in WT nuclei was greater than in *Orb2^R^* nuclei. In addition, a group of nuclei that lacked the Eya protein in *Orb2^R^* flies was observed in the basal part of the brain between the mushroom body calyces (a white oval in Figure 7). 

The antibody against the Csp protein stained the central brain regions enriched in axons and dendrites, similar to published data [41]. The distribution of Csp protein in the WT brain was similar to that in *Orb2^R^* flies, but the signal was higher in *Orb2^R^* flies. The Pyd protein showed a perinuclear localization in neurons of the central brain and optic lobes and was mostly excluded from the neuropile region. We did not observe a significant change in staining pattern in *Orb2^R^* flies, but the overall signal was higher, as in the case of the Csp protein. 

## 4. Discussion

Previous studies have implicated Orb2 in learning and memory retention [7,8,19]. The two Orb2 isoforms, Orb2A and Orb2B, are believed to have different functions in LTM formation. Orb2A is thought to function as a catalyst of proto-prion formation [42]. Upon synaptic activity, Orbr2A primes LTM by undergoing a structural transition of the N-terminal polyQ domain. This enables Orb2A to form heteromeric complexes with the much more abundant Orb2B isoform. The structural transition from a proto-prion to a prion-like structure of the Orb2A polyQ domain generates a self-propagating feedforward loop, which ensures that any newly synthesized or transported Orb2 protein in the local neighborhood of the affected synapse will be captured in these heteromeric complexes. This mechanism provides a basis for long-lasting molecular changes that preserve memory over time. 

Critical to this mechanism proposed for LTM formation, the Orb2 protein must be localized to the synapses responsible for memory formation. Based on the previous studies, several potential ways of targeting Orb2 to synapses could be in play before the events leading to memory formation take place. One of these would be translation of the *orb2* mRNAs in the soma and then either diffusion or active transport of the protein into the neurite. Another mechanism would be on-site translation of the *orb2* mRNAs in proximity to synaptic connections of neurons [43]. In this mechanism, the *orb2* mRNAs would be transported from the soma to neurites in a translationally repressed state. Once on site, they could potentially be stored for future use or translated immediately. In both cases, an active process would be required to convert the translationally repressed mRNA into an active form. Both transport and translational regulation of mRNA typically depend on sequences in the 3′UTR [44]. These UTR sequences are likely to be especially important in neurons, as suggested by the observation that neurons differ from other somatic cell types in that many mRNAs have extended 3′UTRs [45]. 

The *orb2* gene produces multiple mRNAs with different 3′UTRs. Noteworthy, only a single mRNA encodes Orb2A, and this mRNA has a short UTR of only 395 nt without any CPE-binding sequences. The localization and translation of the *orb2A* mRNA are therefore likely to be independent of the Orb2A/B proteins. In contrast, the mRNAs encoding Orb2B have 3′UTRs of 583, 1581, 3829, and 5794 nt. The two larger mRNA isoforms have over 25 CPEs, while the mRNAs with the two shorter UTRs have 2 and 7 CPEs. To determine whether the long UTRs are important for the *orb2* function in the nervous system, we generated a deletion that removed all but the first five CPEs from the 3′UTR (the five CPEs are present in the 1581-nt isoform) [26]. This mutation did not affect the 3′UTR of the mRNA isoform encoding Orb2A. 

Although we did not find reductions in viability or locomotion, striking defects were observed in LTM retention in *orb2^R^* flies. This finding indicates that the 3′UTR of the long *orb2* isoforms plays an important role in LTM. The deletion does not appear to affect *orb2* transcription or mRNA stability because the levels of the *orb2* mRNAs in *orb2^R^* are similar to those in WT. However, the accumulation of the Orb2B protein is altered; i.e., we found that the total amount of the Orb2B protein in *orb2^R^* heads was approximately 2/3 of that in WT heads. Although this reduction is modest, it is conceivable that what is hypothesized to be a key step in memory consolidation, the assembly of heteromultimers between Orb2A and Orb2B, might depend upon the concentration of Orb2B in the vicinity of critical synapses. This idea is supported by our analysis of the distribution of the Orb2B protein in WT and *orb2^R^* neurons. Fractionation experiments indicate that Orb2B levels in the soma and synaptic fractions are close to equal in WT. In *orb2^R^*, the Orb2B levels in the soma are similar to those in WT; however, the protein amount in the synaptic fraction is nearly three times lower. Therefore, the reduction in Orb2B protein levels in *orb2^R^* appears to be largely restricted to the subcellular compartment that is expected to be intimately involved in the formation of LTM dependent on mRNA translation.

Using the RNA IP approach combined with high-throughput sequencing, we obtained a list of 6415 genes whose mRNAs are potential targets for binding and regulation by the Orb2 protein in the *Drosophila* nervous system. Analysis of the list of pulled-down mRNAs indicates that it fits with previously published results. The list is enriched in genes thought to be important for learning and LTM formation [46,47,48,49,50]. Moreover, Orb2 has previously been shown to participate in asymmetric cell division in neuroblasts [17]. Regulators of this process [51,52,53] are significantly enriched in our list of pulled-down mRNAs.

In general, the list is enriched in genes specific for neuroblasts [54,55,56,57,58] and distinct neural subtypes [59,60,61,62,63,64,65], important for neurogenesis [66,67,68,69] and nervous system development [70,71,72,73]. There are also genes that are important for synaptogenesis [74]; synapse assembly, structure, and function [75,76,77,78]; and synaptic plasticity [79] in the list. Genes required for neuron-dendrite morphogenesis [80,81,82], neuronal wiring [83], axon guidance [84,85], axon and dendrite pruning [86,87], and neuronal remodeling [88,89,90,91] are enriched in the list. Interestingly, the list includes multiple genes that have been reported to be important for axonal growth after injury upon Orb overexpression in neurons [92]. mRNAs that have been shown to exhibit specific localization patterns [93] and targets of cytoplasmic polyadenylation [94] were also found among the pulled-down mRNAs.

The list includes genes for proteins interacting with the microtubule cytoskeleton [95,96] and cytoskeletal machinery of neurons during axonal growth and maintenance [97]. At the same time, the list contains many genes encoding transcription factors [98,99], regulators of RNA metabolism [100,101], and components of signal-transduction systems [102,103]. Thus, Orb2 apparently mediates diverse molecular mechanisms of synaptic plasticity, and our list of putative Orb2 targets provides a useful tool for further studies in the field.

We evaluated the influence of the *orb2^R^* mutation on the intracellular distribution of proteins whose mRNAs were identified as Orb2 targets. The Orb2 protein has been shown to bind the *pyd* and *csp* mRNAs, but not the *eya* (found in our RNA IP experiment) mRNAs in CLIP experiments [34]. Although the first two mRNAs are broadly expressed in flies and the cell culture used in CLIP analysis, *eya* expression is limited to a subset of cells, including neurons. The *csp* gene encodes a synaptic vesicle chaperone [104], which is expected to occur in dendrites and synapses. However, Csp is found in the soma as well. The *pyd* protein product is found in different parts of the cell, is involved in cell-to-cell junctions, and interacts with cytoskeletal proteins [105,106]. Finally, *eya* encodes a transcription factor, which is expected to occur in the nucleus (soma) [107]. It should be noted that all mentioned genes carry multiple CPE sequences in their 3′UTRs. We found that the *orb2^R^* mutation changes the intracellular distribution of the Pyd and Csp proteins. The levels of both of these proteins increased in the synaptic fraction in the mutant compared to the WT control. Given the reduction in Orb2B protein in the synaptic fraction, this finding would suggest that Orb2B normally functions to repress translation of the *pyd* and *csp* mRNAs in the synaptic compartment. A different result was observed for *eya*. The accumulation of the Eya protein in the somatic fraction (nuclei) is reduced in the *orb2^R^* mutant, indicating that *orb2B* likely functions as a translational activator of this mRNA in the soma compartment.

## Figures and Tables

**Figure 1 cells-12-00318-f001:**
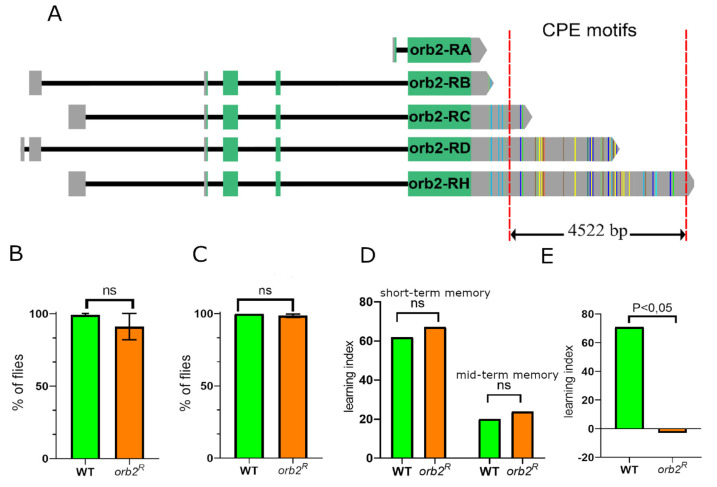
Influence of the *orb2* 3’UTR deletion on fly locomotion and memory formation. (**A**) Schematic representation of *orb2* mRNAs. The vertical-colored lines in the 3′UTR (gray region) show the CPE sequences. Different colors of lines correspond to different CPE sequences (see [26] for details). The red dashed lines show the deleted part of the 3′UTR. (**B**,**C**) The percentage of WT and *orb2^R^* flies that could climb after strong shaking (climbing assay, **B**) and restore the normal body position after intense shaking (vortex assay, **C**). The standard deviation is shown, *t*-test, *n* = 100; ns – nonsignificant (**D**,**E**) Learning indices of WT and *orb2^R^* flies as calculated for short-, mid- (**D**), and long-term (**E**) memory by assessing the suppression of male sexual behavior after interaction with a previously fertilized female. Assessment of differences between learning indices based on calculated courtship indices (two-side randomization test, *p* < 0.05, *n* = 20).

**Figure 2 cells-12-00318-f002:**
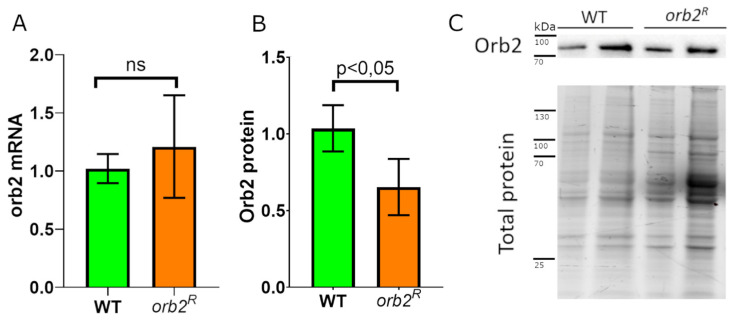
Effect of the 3′UTR deletion on *orb2* gene expression in the brain. (**A**) Relative levels of the *orb2* mRNA in the heads of WT and *orb2^R^* flies as determined by qPCR. ns - nonsignificant. (**B**) Relative amount of the Orb2 protein in the heads of WT and *orb2^R^* flies as detected by semiquantitative Western blot. Error bars show the standard deviation (*t*-test, *n* = 5). (**C**) Western blot analysis of head lysates of WT and *orb2^R^* flies with antibodies against the Orb2 protein (**top**). Total protein staining (**below**) was used for normalization in two-fold serial dilution.

**Figure 3 cells-12-00318-f003:**
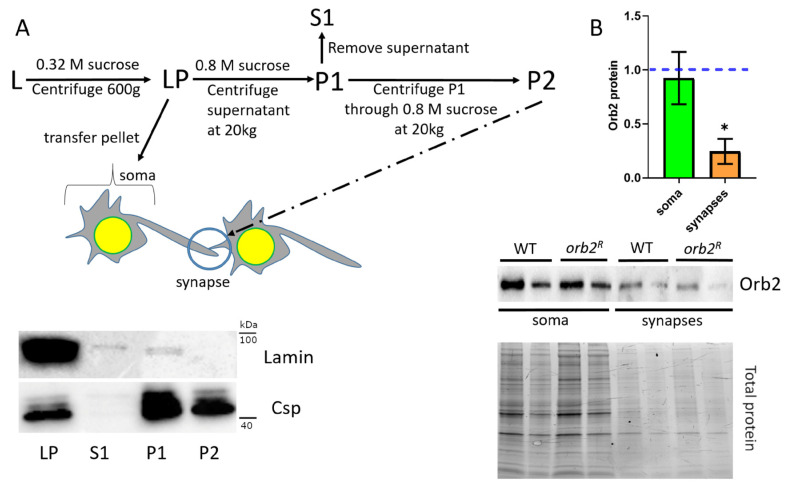
Distribution of the Orb2 protein in neurons. (**A**), top panel: Scheme of fractionation of the brain extract (see text for detail). (**A**), bottom panel: Western blot analysis of the fractions indicated in (**A**) with antibodies against Lamin and Csp. (**B**), top panel: The Orb2 protein contents in synapses and soma in WT and the *orb2^R^* mutant. Error bars show standard deviation (*t*-test, *n* = 10). A blue dashed line displays the relative protein levels in WT flies; bars show the levels in *orb2^R^* flies. An asterisk shows the significant change (*p* < 0.05) in protein levels in *orb2^R^* flies compared with WT flies. (**B**), bottom panel: Western blot hybridization with antibodies against the Orb2 protein. LP (soma) and P2 (synaptic) fractions were analyzed. Each fraction was used in two two-fold dilutions (10 and 5 µL). Total protein staining was used for normalization.

**Figure 4 cells-12-00318-f004:**
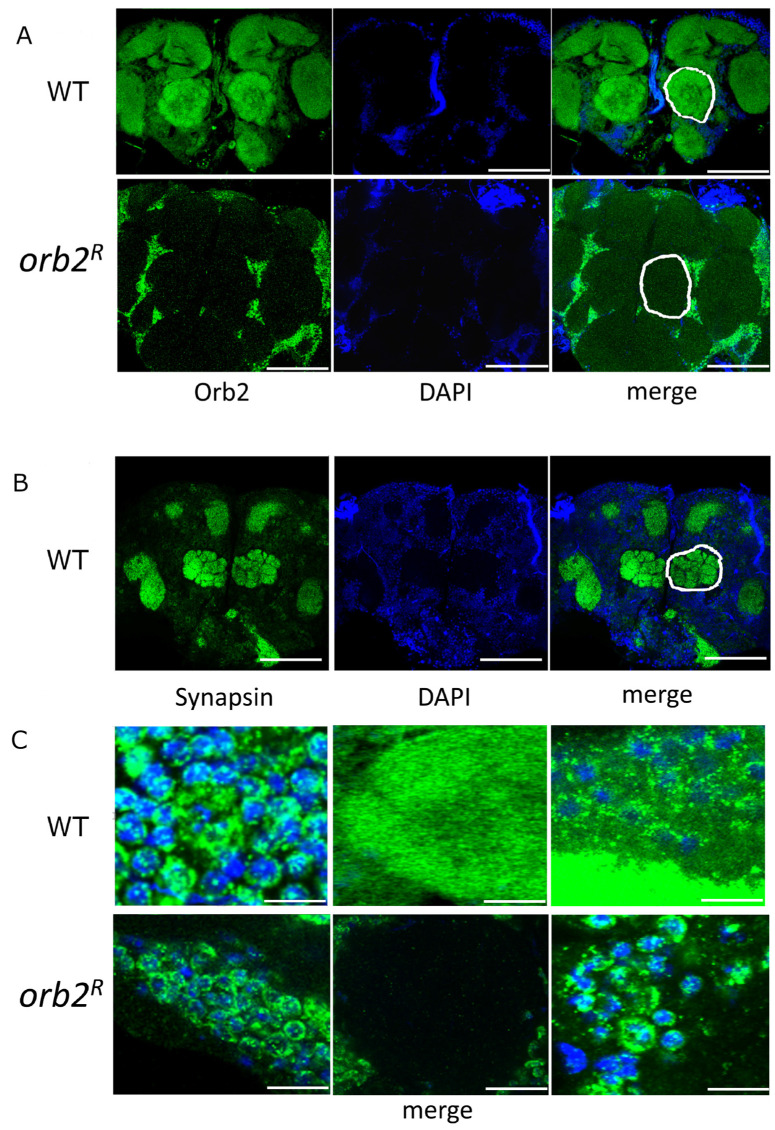
Distribution of the Orb2 protein in the brain of *Drosophila* in WT and *orb2^R^* mutants. (**A**) Staining of the entire brain of *Drosophila* with antibodies against the Orb2 protein (green) and DAPI (blue). The images were obtained with equal exposure. A white line shows the neuropile region. Scale bar, 100 µm. (**B**) Staining of the WT fly brain with anti-Synapsin antibody to visualize the major neuropile regions (one of them is marked with a white line). Scale bar, 100 µm. (**C**) Enlarged images of the neuron soma region (left) and neuropile region (middle) in samples of WT and *Orb2^R^* flies. Merged images are shown. On the right are long-exposure images to visualize soma and neuropil regions in the same image. Whole brain images for these panels are given in the Appendix A. Scale bars for top images are 5, 20, and 5 µm (from left to right). Scale bars for bottom images are 10, 10, and 5 µm (from left to right).

**Figure 5 cells-12-00318-f005:**
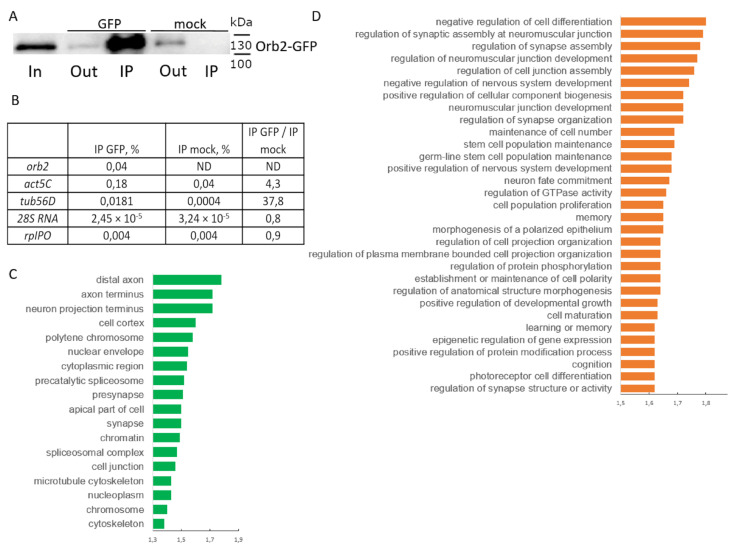
Immunoprecipitation of Orb2 mRNA targets. (**A**) Western blot analysis of samples obtained in the IP procedure with antibodies against GFP. (**B**) The yield (% of input) of several mRNAs obtained in the IP reaction and enrichment of these mRNAs in GFP IP vs. mock IP. (**C**,**D**) Overrepresented GO terms of cell components (**C**) and biological processes (**D**) in genes of the list obtained. Bars represent fold enrichment.

**Figure 6 cells-12-00318-f006:**
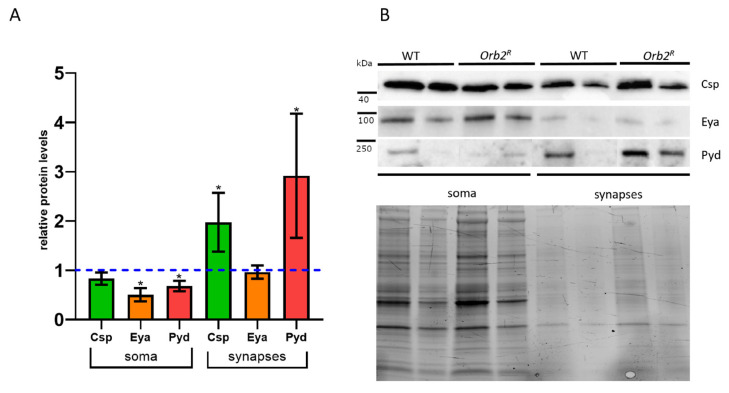
Intracellular distribution of proteins encoded by potential Orb2 mRNA targets. (**A**) The relative protein levels in the soma of neurons and synapses. A blue dashed line displays the protein levels in WT flies; bars show the levels in *Orb2^R^* flies. Samples were obtained from neuron bodies (LP) and the synaptic (P2) fraction. An asterisk shows the significant change (Student *t*-test, *n =* 5, *p* < 0.05) in protein level in *Orb2^R^* flies compared with WT flies. The error bar is the standard deviation. Total protein staining was used for normalization. (**B**) Western blot analysis of the fractions with antibodies against the Csp, Eya, and Pyd proteins. Two replicas of WT and *Orb2^R^* samples are shown.

**Figure 7 cells-12-00318-f007:**
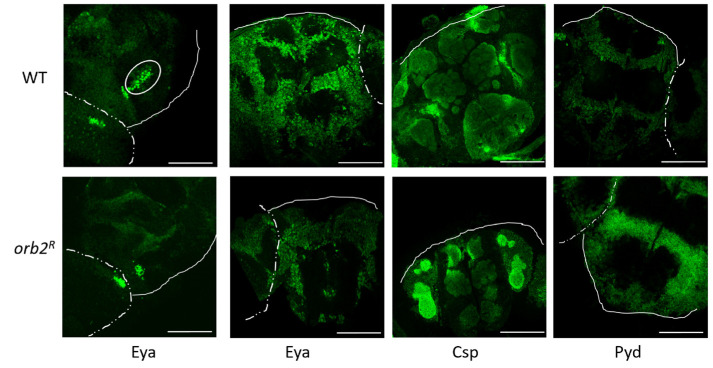
Whole-mount staining of the *Drosophila* brain in WT and *orb2^R^* flies. Staining was performed with antibodies against the Eya, Csp, and Pyd proteins. A dashed line shows the board between the optic lobe and the central brain, and a solid line shows the board of the central brain. White ovals show a group of cells whose nuclei were stained with the antibody against the Eya protein in WT. For each antibody staining, the same exposure was used for samples of WT and *Orb2^R^* flies.

## Data Availability

The data that support the findings of this study are openly available in the BioProject database.

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
