# Peer review of "Long-Term Memory Formation in Drosophila Depends on the 3′UTR of CPEB Gene orb2"

_cells, 2023, doi:10.3390/cells12020318_

Round 1

Reviewer 1 Report

Kozlov et al. examine the function of 3’UTR of Drosophila orb2 in long-term memory formation. To address this, the authors employed a previously described 3’UTR deletion of the orb2 gene, orb2R, that affects the orb2B mRNAs with longer 3’UTRs but not the orb2B mRNAs with short 3’UTRs and isoform encoding for the orb2A. Unlike orb2 mutants, orb2R flies showed normal locomotor activity and short/mid-term memory formation, however defective for the long-term memory of courtship conditioning. Using biochemical and immunofluorescence analysis, the authors show that deletion of the 3’UTR significantly reduces the synaptic localization of the orb2, which further affects the localization of the orb2 mRNA targets. Thus, suggesting the role of orb2 3’UTR in mRNAs localization and translation for long-term memory formation.    

The manuscript is written clearly; however, the authors must provide sufficient methodological details for reproducibility. 

1. Figures: A schematic of orb2 gene coding for different isoforms and orb2R deletion would be helpful for easy readability. 

1a. Provided figures are pixelated. Authors should submit high-resolution figures with clear labeling.

1b. Colors schemes used in the histograms need to be clarified and more accurate. For instance, in Fig. 1A and B, WT and orb2R phenotypes correspond to green and orange colors, respectively. In contrast, the same color schemes correspond to short-and mid-term memory, respectively, but not to the genotypes in Fig. 1C. Also, authors should reconsider using the green and red colors in the same graph.

1c. Error bars are missing in several graphs. Also, the authors should specify what the error bars correspond to – SEM, SD, etc.?

1d. Mention the statistical tests, actual p values, and the number of sets in the figure legend.

1e. Provide a complete title for axis labeling. For instance, in Fig. 1A and 1B: What does the Y-axis title “%” stand for?  

1f. In the western blot, mention the corresponding molecular weight of the proteins. Saturation of the signals in the blots, fails to reflect the accurate protein levels. Authors should reconsider the analysis of these blots.

2. Experimental methods: Authors either provide sufficient information for all methods employed in the manuscript or cite previously published literature. 

2a. Locomotion assays: limited with experimental details for reproducibility.

2b. Behavioral assays: The described steps need more clarity and details. For instance, “Selected males of the strain under study were maintained in individual vials.” – collection steps flies age, the number of flies in a vial, and culture maintenance conditions are missing. How many males and females are used for training? How is memory testing performed? Experimental chambers used for the courtship behaviors? The statistical test used to determine the significance between the groups, etc.

2c. Immunostaining: “….incubation continued for 24 h” and “…. incubation continued for 2 h” – incubation conditions are missing. “After 3X washing…..” specify the solutions used for washing. Include image acquisition parameters. 

2d. Provide catalog numbers for the products used in the methods.

2e. If the yeast-raisin medium is different from commonly used corn-molasses with yeast medium for culturing the fruit flies in the laboratory, more information would be useful. 

3. Authors show a significant reduction in orb2 in the synaptic fraction of orb2R mutants than WT controls. The immunofluorescence analysis fails to support this observation. While the immunofluorescence signal is significantly reduced in orb2R mutants, it is difficult to compare the localization of the orb2 in the CNS, as the authors provide different confocal sections in WT and orb2R brains. Additionally, co-staining with synaptic antibodies, which are well characterized in Drosophila CNS, would be effective.  

4. Immunostaining the whole brain with an antibody (nc82, syn, etc) that enables visualization of different brain regions helps to better interpret the data presented in Fig. 7. To support the claims made in the manuscript, the authors must provide high-resolution images.   

5. Has authors used previously reported orb2 mutants (orb26090orb2ΔA) in their experimental paradigms. These mutants can be appropriate controls for orb2R mutant analysis.  

Minor comments:

1. Cite the corresponding figures in the text. In line 112, Fig. 1 should be Fig. 1A & 1B; in line 150, Fig. 2 should be Fig. 2A, and so on. 

2. Replace orb2R with orb2R (Lines 128, 169)

3. Citation of the previous studies is excluded in several places. For example:

Line 54-56: “Cytoplasmic polyadenylation element (CPE)-binding (CPEB) proteins are a family of translation factors involved in LTM formation. The CPEB proteins bind to U-rich sequences known as CPEs in the 3’UTRs of mRNAs to control their translation.”

Lines 129- 131: “To investigate the effects of orb2R on the learning and memory retention, we used the conditioned courtship suppression paradigm in males, which were tested after interacting with non-receptive females.”

4. Throughout the manuscript, replace 0C with °C.

5. Carefully check the manuscript for English editing and typographical errors.

Author Response

The authors sincerely thank reviewers for a thorough and careful reading of the work and insightful comments. Our answers to first reviewer comments are as follows:

Comment #1

  1. Figures: A schematic of orb2 gene coding for different isoforms and orb2R deletion would be helpful for easy readability.

Reply #1

We have added a scheme of Orb2 mRNAs in figure 1.

Comment #2

1a. Provided figures are pixelated. Authors should submit high-resolution figures with clear labeling.

Reply #2

We have carefully checked the resolution of presented images.

Comment #3

1b. Colors schemes used in the histograms need to be clarified and more accurate. For instance, in Fig. 1A and B, WT and orb2R phenotypes correspond to green and orange colors, respectively. In contrast, the same color schemes correspond to short-and mid-term memory, respectively, but not to the genotypes in Fig. 1C. Also, authors should reconsider using the green and red colors in the same graph.

Reply #3

Please, find updated version of figure 1.

Comment #4

Error bars are missing in several graphs. Also, the authors should specify what the error bars correspond to – SEM, SD, etc.

Reply #4

We carefully checked a presence of error bars in graphs. As a common practice, learning indices are presented without error bars. Please, find examples in the following papers:

Keleman, K., et al., Function of the Drosophila CPEB protein Orb2 in long-term courtship memory. Nat Neurosci, 2007. 10(12): p. 1587-93.

Krüttner, S., et al., Drosophila CPEB Orb2A mediates memory independent of Its RNA-binding domain. Neuron, 2012. 76(2): p. 383-95.

Comment #5

1d. Mention the statistical tests, actual p values, and the number of sets in the figure legend.

Reply #5

We have added the mention information in the figure legends.

Comment #6

1e. Provide a complete title for axis labeling. For instance, in Fig. 1A and 1B: What does the Y-axis title “%” stand for?

Reply #6

We have updated the axis labeling.

Comment #7

1f. In the western blot, mention the corresponding molecular weight of the proteins. Saturation of the signals in the blots, fails to reflect the accurate protein levels. Authors should reconsider the analysis of these blots.

Reply #7

Images of Western blots presented in the figures were exported using Image Lab software with active function “highlight overexposed pixels”. We did not find oversaturated signals. Please, find original figures of Western blots with highlighted artefacts on border of the gels.

Comment #8

  1. Experimental methods: Authors either provide sufficient information for all methods employed in the manuscript or cite previously published literature.

2a. Locomotion assays: limited with experimental details for reproducibility. done

2b. Behavioral assays: The described steps need more clarity and details. For instance, “Selected males of the strain under study were maintained in individual vials.” – collection steps flies age, the number of flies in a vial, and culture maintenance conditions are missing. How many males and females are used for training? How is memory testing performed? Experimental chambers used for the courtship behaviors? The statistical test used to determine the significance between the groups, etc.

Reply #8

We have provided more details about behavioral assays in the text.

Comment #9

2c. Immunostaining: “….incubation continued for 24 h” and “…. incubation continued for 2 h” – incubation conditions are missing. “After 3X washing…..” specify the solutions used for washing. Include image acquisition parameters.

2d. Provide catalog numbers for the products used in the methods.

Reply #9

We have included additional information in the text.

Comment #10

2e. If the yeast-raisin medium is different from commonly used with yeast medium for culturing the fruit flies in the laboratory, more information would be useful.

Reply #10

Medium for Drosophila culturing were similar to commonly used yeast medium. We added description of medium component to the text.

Comment #11

Authors show a significant reduction in orb2 in the synaptic fraction of orb2R mutants than WT controls. The immunofluorescence analysis fails to support this observation. While the immunofluorescence signal is significantly reduced in orb2R mutants, it is difficult to compare the localization of the orb2 in the CNS, as the authors provide different confocal sections in WT and orb2R brains. Additionally, co-staining with synaptic antibodies, which are well characterized in Drosophila CNS, would be effective.

Reply #11

We have updated Figure 4 to show an optical section depicting the described reduction of the Orb2 signal in mutant flies. Also, we have added a separate image showing localization of neuropile marker. Unfortunately, we are not able to perform double staining because all of the antibodies used were raised in mice. Raising of antibodies against Orb2 protein in another species is difficult at this time.

Comment #12

Immunostaining the whole brain with an antibody (nc82, syn, etc) that enables visualization of different brain regions helps to better interpret the data presented in Fig. 7. To support the claims made in the manuscript, the authors must provide high-resolution images.

Reply #12

Indeed, additional markers of brain regions would allow a clearer identification (if available) of changes in protein localization. Unfortunately, we only have mouse antibodies against marker proteins, which makes difficult to perform co-staining. Obtaining antibodies produced in another animal is currently difficult.

The brain regions mentioned in the text were identified using flybrain.org by nuclei localization (DAPI staining, not given in the paper). Moreover, we do not assume specific changes in the protein distribution in individual brain regions based on the data obtained.

Comment #13

Has authors used previously reported orb2 mutants (orb26090, orb2ΔA) in their experimental paradigms. These mutants can be appropriate controls for orb2R mutant analysis.

Reply #13

We did not use the mentioned lines with mutations in the orb2 gene because they affect the coding part of the gene. In our work we have focused on the role of the regulatory regions of orb2 mRNA.

Comment #14

Minor comments:

  1. Cite the corresponding figures in the text. In line 112, Fig. 1 should be Fig. 1A & 1B; in line 150, Fig. 2 should be Fig. 2A, and so on.

Reply #14

We have updated citations of the figures.

Comment #15

  1. Replace orb2R with orb2R (Lines 128, 169).

Reply #15

We have made the mention substitution.

Comment #16

  1. Citation of the previous studies is excluded in several places. For example:

Line 54-56: “Cytoplasmic polyadenylation element (CPE)-binding (CPEB) proteins are a family of translation factors involved in LTM formation. The CPEB proteins bind to U-rich sequences known as CPEs in the 3’UTRs of mRNAs to control their translation.”

Lines 129- 131: “To investigate the effects of orb2R on the learning and memory retention, we used the conditioned courtship suppression paradigm in males, which were tested after interacting with non-receptive females.”

Reply #16

We have added citations of previous studies.

Comment #17

Throughout the manuscript, replace 0C with °C.

Reply #17

The mentioned substitutions have been done.

Reviewer 2 Report

The paper is devoted to analysis of the functional role of three prime untranslated region of a very interesting for neurobiologists gene orb2.  The 3′-UTR often contains regulatory regions  and authors using multiple approaches obtained very interesting results. The paper is well written, English is close to perfect, scientific logic is evident. The results are clear, and the paper can be recommended for publishing.

Minor problems:

Some figures in the file were of low resolution. Hope that final figures will be of adequate resolution.

Author Response

The authors sincerely thank reviewer for a careful reading of the work and positive evaluation. Our answer to the comment is follwoing:

Comment #1

Some figures in the file were of low resolution. Hope that final figures will be of adequate resolution.

Reply #1

We have updated the figures.

Reviewer 3 Report

The manuscript entitled “Long-term memory formation in Drosophila depends on the 2

3’UTR of CPEB gene orb2” by Kozlov et al. studies the function of the CPEB protein Orb2 of Drosophila  in synaptic plasticity. It is during embryogenesisThey observed that the 3’UTR of the orb2 mRNA is important for the proper localization of Orb2 in synapses, facilitating memory formation. This is a well-written manuscript with relevant results about the control of synaptic plasticity. I have only some comments to clarify the interpretation of the results.

Abstract, the introduction of the issue is difficult to understand to me. I suggest to replace the sentence “CPEB proteins can be involved in activating dormant mRNA translation through a polyA-dependent mechanism” with the phrase “CPEB proteins are a family of translation factors involved in LTM formation”

Figure 4. Symbols are not clearly observed. Please, increase the size.

Figure 5. At least in my version, the quality of the text in C and D is difficult to read. Check if it is fine in the original version of the figure.

Methods, when was the heat-shock assay used?

Author Response

The authors sincerely thank reviewer for a careful reading of the work and positive evaluation. Our answer to the comments is following:

Comment #1

Abstract, the introduction of the issue is difficult to understand to me. I suggest to replace the sentence “CPEB proteins can be involved in activating dormant mRNA translation through a polyA-dependent mechanism” with the phrase “CPEB proteins are a family of translation factors involved in LTM formation”

Reply #1

The mentioned phrase has been replaced.

Comment #2

Methods, when was the heat-shock assay used?

Reply #2

We have detailed heat-shock assay in the main text.

Round 2

Reviewer 1 Report

The authors have satisfactorily addressed the reviewer's concerns, except for the figures. Authors need to consider providing high-resolution images for easy readability.

The distribution of Orb2 protein in the fly brain remains unconvincing. In figure 4A, DAPI staining is not visible in the fly brain. High-magnification confocal images are recommended. Alternatively, given the challenges with Orb2b antibody, the authors can consider performing the experiments with transgenic flies - synaptic proteins tagged with fluorescent proteins. 

Author Response

The authors sincerely thank the reviewer for positive evaluation of our work and useful comments. Our answers to reviewer comments are as follows:

Comment #1

Authors need to consider providing high-resolution images for easy readability.

Reply #2

We have increased the resolution of the figures in the text. Moreover, high-resolution images are uploaded separately.

Comment #2

The distribution of Orb2 protein in the fly brain remains unconvincing. In figure 4A, DAPI staining is not visible in the fly brain. High-magnification confocal images are recommended. Alternatively, given the challenges with Orb2b antibody, the authors can consider performing the experiments with transgenic flies - synaptic proteins tagged with fluorescent proteins.

Reply #2.

We replaced the image of the wild-type fly brain and used a preparation with better DAPI staining (Fig. 4A). In addition, we added several images at maximum magnification (Fig. 4C, Fig S1-6). Collectively, these images, taken on different preparations and showing different optical sections, support the conclusion that in mutants there is a significant change in the amount of Orb2 protein in the neurites but not in the neuronal soma.
The proposal to create flies with fluorescently labeled synaptic proteins would require a lot of work. Unfortunately, the format of the response in this journal does not allow such work to be done in the required time frame.

Reviewer 3 Report

The authors appropriately addressed all concerns raised with the initial manuscript submission

I have no more suggestions to make.

Round 3

Reviewer 1 Report

In the revised version, the authors have made significant improvements in the data presentation and provided high-resolution images.